# Emergent Discrete Communication in Semantic Spaces

Mycal Tucker[1], Huao Li[2], Siddharth Agrawal[3], Dana Hughes[3], Katia Sycara[3], Michael Lewis[2], and Julie Shah[1]

1 Massachusetts Institute of Technology
2 University of Pittsburgh
3 Carnegie Mellon University
mycal@mit.edu, hul52@upitt.edu, siddhara@cs.cmu.edu

## Abstract

Neural agents trained in reinforcement learning settings can learn to communicate among themselves via discrete tokens, accomplishing as a team what agents would be unable to do alone. However, the current standard of using one-hot vectors as discrete communication tokens prevents agents from acquiring more desirable aspects of communication such as zero-shot understanding. Inspired by word embedding techniques from natural language processing, we propose neural agent architectures that enables them to communicate via discrete tokens derived from a learned, continuous space. We show in a decision theoretic framework that our technique optimizes communication over a wide range of scenarios, whereas one-hot tokens are only optimal under restrictive assumptions. In self-play experiments, we validate that our trained agents learn to cluster tokens in semantically-meaningful ways, allowing them communicate in noisy environments where other techniques fail. Lastly, we demonstrate both that agents using our method can effectively respond to novel human communication and that humans can understand unlabeled emergent agent communication, outperforming the use of one-hot communication.

## 1 Introduction

A longstanding goal of AI has been to develop agents that can cooperate with other agents or humans to accomplish tasks together. Often, communication is necessary to enable such cooperation; the study of emergent communication has recently shown great success in producing agents that learn to communicate. In reinforcement learning settings, guided only by environment reward, neural agents can learn to communicate by broadcasting numerical vectors to each other [9, 11, 31, 18].

Given this success, and in part inspired by the discrete nature of words in natural language, some researchers have focused on emergent discrete communication by forcing agents to broadcast one-hot vectors [9, 20]. These tokens in effect become a lexicon used by agents. Studying when these tokens are emitted allows researchers to uncover their meanings, as well as to study the broader questions of what environment or agent factors contribute to desirable aspects of learned communication (e.g., compositionality or, in continuous communication settings, zero-shot understanding) [15, 19, 5, 4].

We claim that discretizing messages by constraining them to conform to one-hot vectors fundamentally precludes agents from learning some desirable properties of language. One-hot vectors establish no relationships between tokens because each one-hot vector is orthogonal to and equally far away from all other vectors. Conversely, research from natural language processing and word embeddings has long established the importance of learning representations of discrete words within a continuous, semantic space [29, 32].

35th Conference on Neural Information Processing Systems (NeurIPS 2021).

In this work, we demonstrate the benefit of agents that employ a discrete set of tokens within a continuous space over agents that use the standard practice of communicating via one-hot vectors in discrete emergent communication settings. We present a novel architecture and implementation for learning such communication and provide decision-theoretic analysis of the value of such an approach - the congruence of meaning and form of communications. Simulation experiments confirmed these results: our agents learned an arrangement of tokens that clustered in human-understandable patterns. The arrangement of discrete tokens within the learned communication space produced team performance that was robust to environment noise and enabled agents to effectively utilize novel communication vectors. In human-agent experiments, agents aligned their tokens with natural language embeddings and responded appropriately to novel English phrases. Lastly, we showed that humans capably interpreted unlabeled emergent communication tokens in a reference game.[1]

## 2 Related Work

We propose a technique within emergent communication literature, drawing inspiration from work on word embeddings in natural language processing (NLP) and zero-shot classification.

### 2.1 Emergent Communication

Researchers of emergent communication study techniques to enable agents to learn to communicate among themselves, enabling high task performance in reinforcement learning settings (see [36, 10, 18], among others). These settings, such as reference games or Lewis signalling games [25], are designed such that agents must communicate to perform the task successfully; in "cheap-talk" scenarios, agents often learn successful communication strategies by sending real-valued vectors to each other. We focus on discrete communication emerging among decentralized agents that may communicate by sending one of a finite set of vectors to each other, but may not access other agents' weights or gradients during training or execution [8]. Previously, such discrete communication has often taken the form of one-hot messages: Foerster et al. [9] proposed binary discrete messages, but subsequent works seem to have reverted to one-hot vectors [15, 20, 11, 23, 6, 40].

Even when agents learn to communicate, they often fail to learn a protocol that humans or separately-trained agents can understand; that is, they fail at the "zero-shot" learning problem [12]. To address this gap, several recent works have found properties of environments that encourage zero-shot continuous communication [5, 4]. Other research addresses human-understandable communication in RL settings. Using bottom-up approaches, researchers align emergent communication tokens with human language [22, 26]; Lazaridou et al. [20] even demonstrated that humans can understand agents' tokens in zero-shot settings (e.g., interpreting the agent's token for "dolphin" as referring to a photo of water). Complementary, "top-down" approaches leverage pre-trained task-specific language models for natural language communication [21].

We position our work within the field of emergent discrete communication: we seek to have agents learn to communicate via a finite set of possible messages, without access to a task-specific language model. Unlike prior work, we relax the assumption that discrete tokens must take the form of one-hot vectors and demonstrate that our proposed architecture enables agent utilization of novel communication, and supports human understanding of learned communications.

### 2.2 Natural Language Word Embeddings

While early neural NLP research used one-hot representations of words, most modern techniques rely upon dense word embeddings such as Word2Vec or GLoVE for faster training and greater generalization power [2, 29, 32, 1]. In essence, these embedding techniques learn discrete representations of words within a semantically meaningful space; words representing similar meanings are often embedded in similar locations. We argue that current research in emergent discrete communication that relies upon one-hot tokens is similar to the early one-hot word embedding techniques that suffered from limited generalizability. Instead, we propose that discrete emergent communication should be learned within a continuous, semantically meaningful space, like modern word embeddings.

---

[1]Anonymized code available at https://anonymous.4open.science/r/NeurIPS-protocomms

## 2.3 Zero-Shot Classification

Research in zero and few-shot classification seeks to train classifiers that are able to correctly classify inputs belonging to new classes despite seeing few or no training examples of that class [38, 28, 7]. Often, techniques employ "side-information" to enable high performance, for example using labeled data from a different domain [28] or information from language [35]. We focus on emergent communication rather than classification, but in experiments aligning communication tokens with word embeddings, we similarly exploit side information for zero-shot understanding.

# 3 Technical Approach

## 3.1 Emergent Communication in Multi Agent Reinforcement Learning

We adopt standard methods for training multiple agents in a decentralized partially-observable Markov decision process [3]. The Dec-POMDP is defined by the $(S, A, T, R, O, \Omega, \gamma)$ tuple. $S$ is the set of states; $A_i \forall i \in [1, N]$ are the sets of actions, including communication, for each of $N$ agents; $T : S \times A_1...A_N \longrightarrow S$ is the probabilistic transition between states due to joint actions. We focus on partially-observable settings to encourage communication among agents. $\Omega$ defines the set of possible observations; $O_i : A_1...A_N \times S \longrightarrow \Omega$ maps from joint actions and the state to distributions of observations for each agent. Lastly, $\gamma$ and $R$ define the discount factor and reward function, respectively. The goal is to find the policies for all agents, $\pi_i : o \in \Omega \longrightarrow A_i$, that maximize the expected discounted reward. Multi-agent Dec-POMDPs are well studied [30, 27, 34]; we borrow existing techniques for solving these problems in multi-agent reinforcement learning settings, using an agent architecture we develop.

## 3.2 Decision-Theoretic Noisy Discrete Communication

In this section, we formalize a noisy channel reference game, an instance of a Dec-POMDP. We derive that one-hot vectors are optimal tokens under restrictive assumptions, and relaxation of those assumptions may lead to suboptimality of one-hot communication.

Consider two agents, a speaker and a listener, that communicate via a $c$ dimensional noisy channel, Each episode is composed of two timesteps. In the first timestep, a speaker emits a token, $t$, chosen from a distribution over columns in a matrix of $z$ tokens, $T$; $t \in T_{c \times z}$. In the second timestep, the listener observes a message, $m \in R^c$, a noisy version of $t$, corrupted by additive zero-mean noise. We constrain all tokens to fall within the unit hypercube (i.e., $0 \leq T[i, j] \leq 1; i \in [0, c], j \in [0, z]$), which constrains the relative scales of tokens and noise.

The listener's task is to predict which token the speaker emitted, given the observed noisy message. We assume there exists some reward matrix, $R_{z \times z}$ that specifies the shared reward among agents, where entry $R[i, j]$ corresponds to the reward if the speaker emits $T[i]$ and the listener predicts $T[j]$; standard reward matrices resemble the identity matrix, potentially with some small, positive values off the diagonal for "reasonable" mistakes. From a decision-theoretic framework, the speaker and listener wish to maximize expected reward. We seek to find the optimal set of tokens, $T^*$, that maximizes this expected value, as shown below:

$$T^* = \arg\max_T \sum_{i \in z} P(T[i]) \mathop{\mathbb{E}}_{P(m_i|T[i])} \left[ \max_{j \in z} R[i,j] \frac{P(T[j]|m_i)P(T[j])}{\sum_{k \in z} P(T[k]|m_i)P(T[k])} \right] \tag{1}$$

$$= \arg\max_T \sum_{i \in z} \mathbb{E}_{P(m_i|T[i])} \left[ \frac{P(T[i]|m_i)}{\sum_{k \in z} P(T[k]|m_i)} \right] \qquad \text{Uniform prior;} \quad R = I \tag{2}$$

$$= \arg\min_T \sum_{i \in z} \mathbb{E}_{P(m_i|T[i])} \left[ \sum_{j \in z; j \neq i} P(T[j]|m_i) \right] \tag{3}$$

$$= \arg\max_T \sum_{i \in z} \sum_{j \in z; j \neq i} ||T[i] - T[j]||^2 \qquad \text{Gaussian noise} \tag{4}$$

Equation 1 specifies the expected reward using a rational listener model: taking the expectation over tokens (the outer sum), the listener chooses the token $T[j]$ that maximizes the expected reward given the observed message, $m_i$. Equation 2 follows under assumptions of $z = c$ (an equal number of tokens and communication dimensions), $R = I_{z \times z}$, and a uniform prior over emitted tokens. Equation 3 follows, deriving that the optimal tokens minimize the likelihood of confusion of any pair of tokens. Finally, assuming a Gaussian noise model with a variance matrix equal to $\sigma I$ for some constant $\sigma$, Equation 4 states that the optimal tokens maximize the mean squared euclidean distance between tokens. If tokens are constrained within a unit hypercube, one-hot encodings are an optimal solution. A proof of optimality of one-hot tokens is included in Appendix A.

While one-hot encodings may be optimal in some scenarios, the above analysis also reveals when they may not be: when the cost of errors or prior over tokens are not uniform, or when the number of tokens and communication dimensions are not equal. Examples of suboptimality of one-hot tokens when these assumptions are relaxed are included in Appendix A. Thus, although many academic experiments respect these assumptions, a more powerful framework for learning tokens is needed in the general case.

### 3.3   Discrete Prototype Communication

We present our method for prototype-based communication to address the limitations of one-hot communication by endowing agents with learnable tokens. We create neural agents with a communication head instantiated as a multi-layer perceptron with a penultimate softmax layer (where we use the gumbel softmax trick [14]), which then is multiplied by a learnable matrix, $T_{z \times c}$, for the final output. More formally, an agent, parametrized by weights $\Theta$ and $T$, produces a communication token $t$ according to Equation 5.

$$
\begin{aligned}
\pi &= f_\theta(O) \\
d &= \texttt{one\_hot}(\arg\max_i [g_i + \log \pi_i]) \\
t &= d \times T_{z \times c}
\end{aligned}
\tag{5}
$$

Equation 5 states that agents produce logits, $\pi$, from an observation, $O$, convert the logits to a one-hot vector, $d$, by using sampled $g_i$ drawn i.i.d. from a Gumbel(0, 1) distribution, and then multiply $d$ by $T$ to produce the token.[2] Although the argmax operation is not differentiable, we use it in the forward pass (that is, actually using "hard" argmax) but use the gumbel softmax operation for calculating gradients, used in the backwards pass. We refer readers to Jang et al. [14] for more details on the gumbel softmax approximation.

We note that our technique introduces an additional set of learnable parameters compared to one-hot communicative agents, which would merely output $d$. This increased flexibility may enable improved performance but may also increase learning difficulty; experiment results, reported in Sections 5 and 6, indicated net benefits derived from our technique.

### 3.4   Opportunities for Behavior Shaping

The previous section illustrates how $T$ may be learned using traditional reinforcement learning techniques; using a policy gradient method, the tokens will evolve during training to enable high task performance. While this emergent behavior is desirable for its flexibility, we also examined two ways of constraining learned behavior: "lexicon-setting" and grounding.

The first constraint, which we call "lexicon-setting," consisted of manually specifying a non-trainable $T$. During training, gradients passed through $T$ but did not update its elements. This allowed a designer to specify the set of tokens agents used to communicate, in other words setting the "lexicon."

The second constraint consisted of teaching grounding via a small set of supervised data. Even with hand-specified tokens, it is unclear *a priori* what meanings agents will associate with tokens. To enable human-agent communication, we require a common understanding, or grounding, of tokens.

---

[2]In experiments, we constrained the tokens to the unit hypercube by passing $t$ through a sigmoid activation to enable fair comparisons to one-hot encodings, but tokens need not generally obey such constraints.

Inspired by prior art in learning social conventions in RL settings [24, 37, 26], we used a small set of data mapping observations to desired communications. The gradient of the supervised loss (mean squared error of predicted communication vs. desired communication) was added to the policy loss from the RL environment, leading the agent to use communication that matched the grounding data and enabled high task performance.

# 4 Experiment Preliminaries

## 4.1 Baselines

We compared our technique to two emergent communication baselines: continuous and one-hot communication. The continuous communication method (henceforth, *cont*) allowed agents to output real-valued vectors in the communication space, $R^c$, bounded along each dimension within $(-1, 1)$ by using a `tanh` activation. The discrete one-hot baseline (henceforth, *one-hot*) allowed agents to output a one-hot vector within the communication space using the gumbel softmax trick. By definition, this constrained *one-hot* agents to output one of $c$ tokens.

All techniques were trained using the multi-agent deep deterministic policy gradient (MADDPG) method proposed by Lowe et al. [27], a common policy-gradient method that updates neural network weights to maximize the expected discounted reward.

## 4.2 Hypotheses

Throughout our experiments, we assessed trained agents by measuring environment-specific reward under different scenarios. First, we measured self-play reward for agents trained together, as the environment reward they received during 500 evaluation episodes. Second, we measured aspects of zero-shot behavior and performance in a reference game (introduced in Section 6). Measurements included a listener agent's accuracy in selecting a target image when using novel communication generated by humans, as well as human participant accuracy in identifying a target image based on unlabeled agent communication.

Using these metrics, we formulated the following hypotheses:

- **H1:** In noisy environments, prototype-based communication will enable agents to achieve higher self-play reward than one-hot tokens.

- **H2:** Prototype-based agents, trained with word embeddings as tokens, will identify target images in reference games based on novel, human-generated communication, whereas one-hot encodings will fail to outperform random chance.

- **H3:** Human participants will accurately identify target images in reference games based on tokens learned by prototype-based agents.

Our hypotheses were informed by prototype-based communication design, which we expected to cluster tokens with similar meanings, thereby providing robustness in noisy environments and benefiting human understanding of the communication space. We expected this improved semantic understanding of the communication space to bolster zero-shot understanding of novel communication.

# 5 Agent Self-Play Experiments

We first evaluated our technique in particle-world environments, inspired by the environments introduced by Lowe et al. [27] and Mordatch and Abbeel [31]. Agents were trained and tested in fixed teams; thus, these experiments were used to measure robustness to noise and assess characteristics of learned tokens. Depictions of our environments are shown in Figure 1. Full details of the environments are included in Appendix B.

In the first environment, *triangle*, speaker and listener agents were spawned at the origin, and three landmarks were located at the points of an equilateral triangle. In each episode, a landmark was designated as the target, and noise was drawn from a $c = 9$ dimensional zero-mean Gaussian with $\Sigma = 0.9 I_c$. Only the speaker observed the position of the target; thus, to maximize the reward

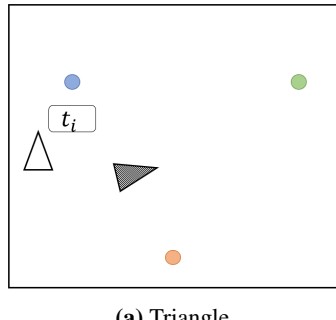

**(a)** Triangle

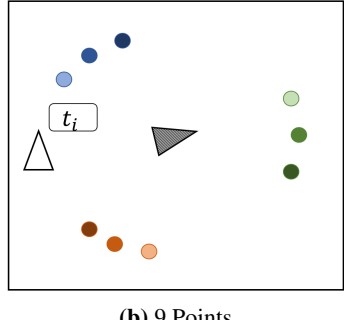

**(b)** 9 Points

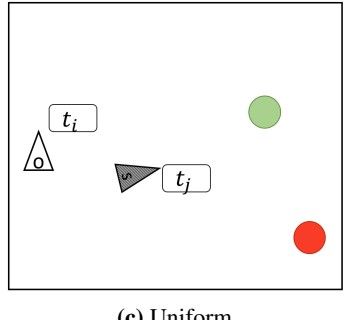

**(c)** Uniform

**Figure 1:** The particle environments used for agent-only testing. In the triangle and 9 Points environments, the landmarks (colored circles) were in fixed locations; a target was chosen among the possible landmarks. In the uniform environment, the landmarks were in locations chosen uniformly at random within the game.

**Table 1:** Median (standard error) self-play reward over 5 runs. Enabling learnable tokens endowed our agents with greater robustness to environmental noise.

| Env. | Dist. | $\sigma$ | c | Rewards | |
|------|-------|----------|---|---------|---|
| | | | | Proto | One Hot |
| Tri. | Unif. | 0.9 | 9 | **-124** (12) | -201 (10) |
| | Skew. | 0.9 | 9 | **-100** (4) | -136 (9) |
| 9-P. | Unif. | 0.0 | 3 | **-60** (7) | -96 (10) |
| | Unif | 0.0 | 9 | -65 (2) | -66 (6) |
| | Unif | 0.5 | 9 | **-100** (10) | -132 (2) |

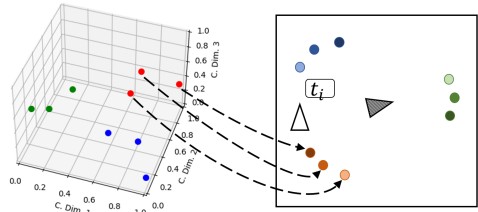

**Figure 2:** The learned prototypes (left) reflect structure in the 9 points environment (right), with similar prototypes encoding similar locations. Prototypes taken from a $c = 3$; $z = 9$; $\sigma = 0.3$ experiment conducted for visualization.

(negative distance from the listener to the target), the speaker must communicate to the listener, via a 9D vector, and the listener must move to the target.

We investigated the effect of different priors over tokens by changing the distribution over which landmark was chosen as the target: in one setting, targets were drawn uniformly, but in the other, targets were drawn from a skewed categorical distribution of $[0.49, 0.49, 0.02]$. Three learnable prototypes were defined for our technique ($z = 3$). Note that this meant that our prototype technique had fewer tokens than the 9 for *one-hot*: we selected $c = 9$ to study scenarios in which $z < c$.

In our second environment, *9 points*, speaker and listener agents were spawned at the origin, with the speaker observing the location of the target and the listener observing its own state. Reward was similarly defined as the negative distance squared from the listener to the target. 9 landmarks were spawned in fixed locations, depicted in Figure 1b. This arrangement of landmarks produced a non-identity reward matrix: the listener confusing two nearby landmarks was less costly than confusing two landmarks that were further apart. We used the *9 points* environment in 3 settings: two settings with $\sigma = 0$ and $c = 3$ or $c = 9$, and one noisy setting with $\sigma = 0.5$ and $c = 9$. In *9 points*, our technique used 9 prototypes. These settings allowed us to study cases when $z \geq c$. Results from the first two environments, assessing robustness to noise, are shown in Table 1.

The *triangle* environment results reveal two benefits of our technique over *one-hot*: robustness to noise when $z < c$ and the ability to exploit a non-uniform prior. First, when there were fewer tokens than communication dimensions, agents used the additional dimensions for robust communication, as shown in the top row of Table 1. The three prototypes spread into distant corners of the 9-dimensional unit hypercube, creating a larger distance between tokens than one-hot encodings. Next, the second row shows that our technique was able to benefit from the non-uniform prior over landmarks by yielding greater robustness for tokens denoting the more likely landmarks. The one-hot technique, in contrast, learned to always go to the location between the two most likely landmarks.

The *9 points* environment enabled us to study two other attributes of learned communication tokens: the benefits of greater expressivity in a channel-limited domain ($z > c$) and clustering behavior due

**Table 2:** Median (std. error) reward over 5 runs in *Uniform*. Continuous communication outperformed both discrete methods, which converged to similar discretization behaviors of the continuous space.

| Method | Reward |
|--------|--------|
| Cont. | **-97.8** (3.8) |
| Proto | -113.6 (3.8) |
| One Hot | -111.8 (3.4) |

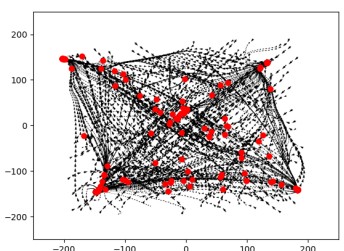

**Figure 3:** Tracing seeker paths (black lines) and final locations (red dots) revealed the discretization imposed by prototypes.

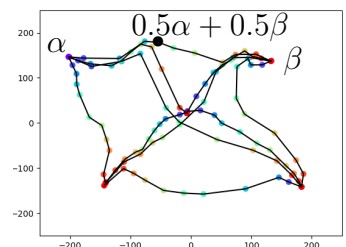

**Figure 4:** Mean destination of the seeker agent when given pairwise interpolations of prototypes from 0% (red) to 100% (purple).

to a non-identity reward matrix when $z = c$. First, we confirmed in the third row of Table 1 that, in an environment with $c = 3$ but 9 landmarks, *one-hot* lacked the expressivity to uniquely denote each target; our technique decoupled the number of tokens and the communication dimension, allowing 9 discrete tokens in a 3D communication channel. The baseline of $z = c = 9$, in the fourth row, confirmed that the 9 prototypes in 3D achieved the same reward as 9 prototypes in 9D or 9 one-hot tokens. Lastly, in the fifth row, we observed that increasing environment noise led to a graceful degradation of performance for prototype-based agents, but worse failures for *one-hot*. Inspection of the learned prototypes reveals why: tokens denoting landmarks in the same cluster of 3 landmarks converged into clusters in the communication space, allowing the clusters to separate from each other, as shown in Figure 2. Thus, intra-cluster errors became more likely, but inter-cluster error likelihood decreased. In fact, the mean reward of -100 achieved in the noisy environment is similar to the reward of -96 achieved by the 3D one-hot communication method: both of these techniques devolved to the behavior of treating clusters as single points.

In our final environment, *uniform*, a seeker and observer agent were spawned at random locations in the world; two landmarks (target and distractor) were also spawned at locations drawn from a uniform distribution over the world bounds. The observer agent observed the location of both landmarks; the seeker agent observed its own position and the id of the target (i.e., the first or second landmark). Each agent communicated to the other at each timestep. To achieve high reward, designated as the negative distance from the seeker to the target, both agents had to communicate: the seeker telling the observer which landmark was the target, and the observer responding with the location of the target. This environment, unlike the prior two, allowed us to assess how agents used a discrete vocabulary to communicate about continuous data. Prototype and *cont* agents communicated with $c = 3$; *one-hot* used $c = 5$, as prototype agents used $z = 5$.

In the *uniform* environment, we found that our method discretized continuous data in a reasonable way and that the emergent tokens enabled some measure of interpolation between representations. First, our method and *one-hot* performed similarly to each other and worse than *cont* (Table 2). Given that the observer agent had to communicate about continuous values (the target's location) with a finite, discrete vocabulary, this is unsurprising. Plotting trajectories of a seeker agent over 200 episodes, the discrete pattern of locations emerges (Figure 3).

Lastly, we tested prototype-based agents with interpolations of learned tokens by overriding the observer agent's communication with pairwise interpolations of all tokens at 10% intervals. An example of the resulting behavior is plotted in Figure 4. (We note that interpolation for continuous agents was impossible because of the lack of discrete tokens, and *one-hot* precludes communication other than one-hot vectors.) Interestingly, we found that prototype-based seeker agents often responded to the interpolation of tokens with interpolation of behavior: for example, when observing the mean of two tokens, $\alpha$ and $\beta$, the seeker agent went to the mean of the locations it would have gone to when observing just $\alpha$ or just $\beta$. These initial findings motivated work in the next section on zero-shot understanding of natural language.

Together, these three experiments provided support for our first hypothesis, **H1**. Our technique enabled agents to learn tokens that provided greater robustness to noise in the *triangle* and *9 points* environments by separating and clustering tokens, confirmed visually in Figure 2.

**Table 3:** CIFAR10 zero-shot reference game mean reward (std. error). Prototype-based models were the only discrete communication method that enabled better-than-random zero-shot understanding.

| Method | Label In | Label Out | Self In | Self Out | AMT In | AMT Out |
|---|---|---|---|---|---|---|
| Cont (BERT) | 91% (0.7) | 55% (1.0) | 91% (0.7) | 68% (1.0) | 73% (0.4) | 53% (0.6) |
| Proto (BERT) | 93% (0.7) | 58% (1.0) | 93% (0.7) | 66% (0.9) | 70% (0.4) | 56% (0.6) |
| Proto (Hand) | 75% (1.1) | 62% (1.2) | 80% (0.6) | 58% (1.0) | | |
| One-hot | 96% (0.2) | 50% (0.8) | 96% (0.2) | 69% (1.0) | | |

## 6 Human-Agent Experiments

Lastly, we tested how well our technique supported aspects of human-agent interaction in zero-shot settings. Results from the prior section already indicated that prototype-based agents learned a sort of semantic communication space: in *9-points*, for example, similar target locations were encoded via similar prototypes, and in *uniform*, the seeker agent responded to interpolations of prototypes with interpolations of behavior. In order to enable human-agent interaction in experiments in this section, we aligned parts of human and emergent communication. We found that this alignment, in conjunction with semantically-meaningful communication spaces, supported two forms of zero-shot understanding. First, agents were able to understand novel human communication at test time (**H2**). Second, humans could understand emergent communication embeddings that agents learned (**H3**).

We conducted these experiments via a reference game, a standard type of environment for emergent communication [12, 5]. In our environment, a speaker observed an image, drawn from the CIFAR10 dataset, and emitted a communication vector [16]. A listener agent observed the original image and a "distractor" image that belonged to a different class, as well as the communication emitted by the speaker. The listener predicted in the second, final timestep which of the two images was the "target" observed by the speaker. Reward was defined via a matrix, $R_{10 \times 10}$, where $R[i, j]$ set the shared reward if the target image belonged to class $i$ and the listener agent predicted an image of class $j$. We penalized some errors more than others according to high-level groupings of classes: correct predictions got reward 1, confusion within the set of 6 animal classes or 4 vehicle classes got 0.75, and predictions that belonged to the wrong group got 0. We transformed this basic reference game into two zero-shot experiments.

### 6.1 Zero-Shot Agent Understanding

In our first zero-shot experiment, we assessed agents' ability to understand novel communication by withholding two image classes (one animal and one vehicle) from training. We trained 5 teams from scratch for each of the 24 possible animal-vehicle excluded combinations, resulting in 120 teams. All agents were trained in environments with $\sigma = 0.05$; *cont* and prototype agents used $c = 3$, prototype agents used $z = 10$, and *one-hot* used $c = 10$.

To align human and agent communication, we augmented self-play training with 100 supervised examples of image-communication pairs. These data were interleaved with self-play in training agents, biasing them to match the desired policy, as done in prior social convention literature [24, 37, 26]. For *cont* and prototype agents, we used a BERT-like model to create communication by embedding the class label names [33]; *one-hot* used the classification label. An additional set of prototype agents was trained with hand-designed word embeddings that clustered vehicle and animal embeddings. See Appendix E for details on embeddings and image feature extraction. For the prototype agents, we used the lexicon-setting approach from Section 3.4 to set $T$ equal to the label embeddings.

During evaluation, we measured listener prediction accuracy over 500 trials when using different types of communication or images. In "in-distribution" experiments, we evaluated models using the 8 image classes seen during training; in "out-of-distribution" experiments, we used only the 2 image classes never seen in training. Furthermore, we tested three types of communication vectors: those produced by the trained speaker agent (Self), by embedding the class label (Label), or by using embeddings of captions generated by human annotators on Amazon Mechanical Turk (AMT). Details of the user study for generating AMT descriptions are included in Appendix F; in essence, participants were asked to provide descriptions of images, replicating the role of the speaker agent.

Results, shown in Table 3, supported hypothesis **H2**: prototype listener agents supported two types of zero-shot communication. AMT In experiments reflected one type of zero-shot understanding by forcing listeners to interpret novel communication at test time. Despite training with a small, finite vocabulary, prototype listeners performed almost on par with *cont* agents. Conversely, one-hot encoding precludes embedding novel communication. AMT Out and Label Out results reflected a second type of zero-shot understanding: listeners simultaneously observed novel images and novel communication. In these settings, our method outperformed *cont* and the random performance of *one-hot*. By testing prototype agents trained using BERT or hand-generated embeddings, we found that different embedding methods played an important role in agent performance. Lastly, in ablation studies, we confirmed that both the reward matrix and environment noise contributed to agents' zero-shot understanding (Appendix D).

### 6.2  Zero-Shot Human Understanding of Tokens

In our final experiment, we studied whether humans could understand agent communication in zero-shot settings, is essence treating humans as listeners in the reference game. We trained prototype agents in the standard reference game, communicating in 2 dimensions with all 10 classes and no explicit guidance on the tokens. See Appendix C for full training details. In evaluation, we showed participants 8 of the 10 learned tokens, annotated with English class labels. Finally, we plotted one of the held-out tokens with no label and showed participants two images, one for each of the two held out classes, and asked participants to select the image they thought the new communication most likely referred to. We recorded participants' selection accuracies among animal-vehicle image pairs for 5 prototype-based models trained from scratch, and a 2D PCA visualization of BERT-based embeddings of labels.

Averaged across the 5 prototype models, with an average of 300 responses per model, participants selected the correct image 70% of the time (standard error of 5%). This compares favorably to the 66% achieved using embeddings derived from the BERT-based model ($\chi^2(1, N = 2093) = 3.21, p = .073$). While prior art has studied how humans interpret labeled agent tokens in new contexts (e.g., interpreting a one-hot token for "dolphin" as referring to a photo of water in Lazaridou et al. [20]), we believe this is the first demonstration of human participants understanding novel tokens. Ultimately, these results provided support for our final hypothesis, **H3**.

## 7  Contributions

In this work, we proposed a technique to address a current gap in existing discrete emergent communication literature. Prior work in this domain relies upon one-hot tokens for communication, but insights from NLP, as well as decision-theoretic analysis, demonstrate why such an encoding scheme is inadequate in many settings. Using a technique we developed to enable learnable, discrete tokens, we demonstrated benefits in self-play, including increased robustness to environment noise. In human-subject experiments, we showed improvements over the random performance of prior art in zero-shot understanding of human communication. Lastly, we demonstrated that prototype agents learned emergent communication tokens that were human-interepretable and semantically meaningful. Overall our experiments provided support that our prototype-based communication method produced semantically meaningful communication spaces in noisy environments with non-identity reward matrices, which in turn enabled zero-shot understanding of communication.

Our work provides a first step towards connecting theories of emergent communication and word embeddings. While we have taken a step in this direction through our proposed technique, future work leveraging insights from discrete representation learning could likely further improve upon our results. In addition, many of the techniques used to examine word embeddings could be used to analyze emergent communication tokens. Specifically, we note that prior literature has found reinforced biases in word embeddings; similar care of encoding desirable properties in emergent tokens must be taken. Lastly, further experiments examining environmental factors that contribute to zero-shot understanding of discrete communication could yield important insight into effective agent training and language evolution.

## Acknowledgments and Disclosure of Funding

We thank Jacob Andreas for early discussions about embeddings in NLP, and the NeurIPS reviewers for their careful feedback. This work has been funded by ARL award W911NF-19-2-0146.

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
