# A  Appendix: Optimality of One-Hot Encodings

We present a brief proof about the local optimality of one-hot encodings in the decision-theoretic framework presented in Section 3.2. We seek to prove that, under assumptions of an identity reward matrix, tokens constrained to a unit hypercube, and gaussian additive noise, one-hot tokens are an optimally robust communication strategy. We only seek to prove local optimality, as one many trivially generate multiple, equally optimal tokens by, for example, flipping all bits.

The following derivation uses Karush–Kuhn–Tucker (KKT) conditions, a generalization of Lagrange multipliers [17].

**Theorem A.1** (Optimality of One-Hot Encodings). *A matrix $T_{c \times c}$, constrained to the unit hypercube, locally maximizes $\sum_{i \in [1,c]} \sum_{j \in [1,c]; j \neq i} (T[i] - T[j])^\top (T[i] - T[j])$ using one-hot encodings.*

*Proof.* We maximize the function, subject to constraints. We denote the $i^{th}$ row of matrix $T$ as $T_i$, and the $j^{th}$ element of the $i^{th}$ row as $T_{ij}$.

$$T^* = \arg\max_{T} \sum_{i \in [1,c]} \sum_{j \in [1,c]; j \neq i} (T_i - T_j)^\top (T_i - T_j) \qquad s.t. \quad 0 \leq T_{ij} \leq 1 \forall i, j \in [1,c] \quad (6)$$

$$\text{We define KKT terms, using } \mu_i \text{ and } \lambda_i \text{ for constraints. } \mu_{ij}, \lambda_{ij} \geq 0 \forall i, j \in [1,c] \quad (7)$$

$$\mathcal{L} = \sum_{i \in [1,c]} \sum_{j \in [1,c]; j \neq i} \left[ (T_i - T_j)^\top (T_i - T_j) \right] - \sum_{i \in [1,c]} \left[ \vec{\mu_i}(T_i - \vec{1}) - \vec{\lambda_i} T_i \right] \quad (8)$$

$$\frac{\partial \mathcal{L}}{\partial \vec{\mu_i}} = -T_i + \vec{1} = \vec{0} \quad (9)$$

$$T_{ij} = 1 \quad \text{if} \quad \mu_{ij} \neq 0 \quad (10)$$

$$\frac{\partial \mathcal{L}}{\partial \vec{\lambda_i}} = T_i = \vec{0} \quad (11)$$

$$T_{ij} = 0 \quad \text{if} \quad \lambda_{ij} \neq 0 \quad (12)$$

$$\frac{\partial \mathcal{L}}{\partial T_i} = \sum_{j \in [1,c]; j \neq i} \left[ \frac{\partial \|T_i\|^2}{\partial T_i} - \frac{\partial T_i^\top T_j}{\partial T_i} - \frac{\partial T_j^\top T_i}{\partial T_i} + \frac{\partial \|T_j\|^2}{\partial T_i} \right] - \vec{\mu_i} + \vec{\lambda_i} = \vec{0} \quad (13)$$

$$(14)$$

We seek to show that one-hot vectors are an optimum, so we now show that one-hot vectors indeed respect the constraints and set the derivatives to zero. By assuming one-hot vectors, we know that $T_i^\top T_j = 0 \forall i \neq j$.

$$\frac{\partial \mathcal{L}}{\partial T_i} = \sum_{j \in [1,c]; j \neq i} \left[ \frac{\partial \|T_i\|^2}{\partial T_i} \right] - \vec{\mu_i} + \vec{\lambda_i} = \vec{0} \quad (15)$$

$$\sum_{j \in [1,c]; j \neq i} [2T_i] - \vec{\mu_i} + \vec{\lambda_i} = \vec{0} \quad (16)$$

$$2(c-1)T_i - \vec{\mu_i} + \vec{\lambda_i} = \vec{0} \quad (17)$$

$$(18)$$

We now consider each element of $T_{ij}$, again assuming that $T_i$ is a one-hot vector.

$$\text{If} \quad T_{ij} = 0 \tag{19}$$
$$\rightarrow 2(c-1)T_{ij} = 0 \qquad\qquad\qquad \text{Because } T_{ij} = 0 \tag{20}$$
$$\rightarrow \mu_{ij} = 0 \qquad\qquad \text{Because the constraint is not active} \tag{21}$$
$$\rightarrow \lambda_{ij} = 0 \qquad\qquad \text{Can set } \lambda_{ij} \text{ to satisfy equation} \tag{22}$$
$$\text{If} \quad T_{ij} = 1 \tag{23}$$
$$\rightarrow \lambda_{ij} = 0 \qquad\qquad \text{Because the constraint is not active} \tag{24}$$
$$\rightarrow 2(c-1)T_{ij} - \mu_{ij} = 0 \qquad \text{Solve with } \mu_{ij} = 2(c-1) \text{ to satisfy equation} \tag{25}$$

Thus, we have shown that one-hot vectors in the unit hypercube are a solution to the constrained optimization problem, indicating that it is a local optimum.

$\square$

One may also prove that one-hot encodings may not be optimal when the assumptions of an identity reward matrix or a uniform prior over tokens are relaxed. These proofs follow from counter-examples: we seek merely to show that there exist other tokens that achieve a higher expected reward. We provide two such examples below for the $z = c = 3$ case, calculating the expected reward for one-hot tokens and alternative tokens we propose, which we show achieve higher mean reward.

### A.1 Uniform prior; non identity reward

$$R = \begin{pmatrix} 1 & 1 & 0 \\ 1 & 1 & 0 \\ 0 & 0 & 1 \end{pmatrix} \tag{26}$$

$$T_{1hot} = \begin{pmatrix} 1 & 0 & 0 \\ 0 & 1 & 0 \\ 0 & 0 & 1 \end{pmatrix} \tag{27}$$

$$T_{alt} = \begin{pmatrix} 1 & 1 & 0 \\ 1 & 1 & 0 \\ 0 & 0 & 1 \end{pmatrix} \tag{28}$$

In this case, there is a reward of 1 even if tokens 0 and 1 are confused. Making those two tokens identical (in $T_{alt}$) therefore incurs no penalty and instead decreases the likelihood of confusion with token 2, thus increasing expected reward compared to using $T_{1hot}$. Running 100,000 simulations of this scenario, with $\sigma = 0.5$, shows that one-hot tokens achieve a mean reward of 0.91, whereas $T_{alt}$ achieves a reward of 0.96.

### A.2 Non-uniform prior; identity reward

$$P = (0.499 \quad 0.499 \quad 0.002) \tag{29}$$

$$T_{1hot} = \begin{pmatrix} 1 & 0 & 0 \\ 0 & 1 & 0 \\ 0 & 0 & 1 \end{pmatrix} \tag{30}$$

$$T_{alt} = \begin{pmatrix} 0 & 0 & 0 \\ 1 & 1 & 1 \\ 0.5 & 0.5 & 0.5 \end{pmatrix} \tag{31}$$

$$\tag{32}$$

Although $R = I$, because tokens 0 and 1 are far more likely than token 2, setting tokens $T$ to maximally distinguish between the first two tokens becomes more important than forcing token 2 to be far from other tokens. Simulations with $\sigma = 0.5$ show that one-hot achieves reward of 0.92 but $T_{alt}$ achieves 0.96.

Together, these two examples prove that when $R \neq I$ or there is a non-uniform prior over tokens, one-hot encodings may not be optimal.

# B   Appendix: Particle Environment Details

The *triangle* environment spawned the speaker and listener agents at the origin. Landmarks were distributed at distance 200 from the origin, every $\frac{\pi}{3}$ radians, starting at 0. The listener particle had a mass of 0.1; its force actions were bounded via a `tanh` operation; force was integrated to change velocity and position via a double-integrator physics engine with a damping coefficient of 0.9. In order to preserve the Markov property of planning, the listener agent observed its own 2D position and 2D velocity. Episodes lasted 50 timesteps.

The *9-points* environment used the same physics engine as *triangle* and episodes also lasted 50 timesteps. Landmarks were located at distance 200 from the origin, with landmark $i$ at angle $\frac{\pi}{3} + 0.3\lfloor\frac{i}{3}\rfloor$ radians. We found that 0.3 was large enough to encourage unique tokens for each landmark, but small enough that clusters were still apparent.

The *uniform* environment used the same physics engine, with both agents' masses as 0.1, but episodes lasted 100 timesteps. We found that additional time enabled more convergent behaviors: that is, agents had to learn to reach a location and stop for maximal reward, as opposed to traveling at full speed in a fixed direction. Landmarks and agents were spawned at locations drawn uniformly at random from the square described by $(-200, -200)$ to $(200, 200)$.

# C   Appendix: Training Details and Hyperparameters

This section specifies the training hyperparameters used in all experiments. When conducting multiple trials for a given environment, each trial was conducted by setting the random seed equal to the zero-indexed trial number.

All agents were trained using an implementation of MADDPG [27], using an existing MIT-licensed PyTorch implementation of the algorithm as an initial development point [13]. When experimenting with environmental noise, all agents were trained using identical noise models and evaluated with zero noise. We used an Adam optimizer with default parameters except for the learning rate, which we set to 0.01. Unless otherwise noted, we set $\tau = 0.01$ – the degree of soft updating of the target networks used in MADDPG.

All agents used a three-layer MLP with ReLU activations and hidden dimension 64 as the base of their policy, with communication or action heads transforming the base output to the desirable form via two layers. Continuous outputs were bounded by a `tanh` activation. Tokens used by prototype agents were bounded to the unit hypercube for fair comparisons with one-hot tokens by passing tokens through a sigmoid activation. In agent-only experiments, we used temperature 0.1; in human-agent experiments, we used temperature 1.0.

All training times were calculated on a desktop computer with an NVIDIA GeForce RTX-2080 graphics card and 16 Intel-i9 CPUs. We reported training times for a single team - total training time for a suite of experiments is calculated by multiplying the numbers of methods, random seeds, and environment settings. In total, all experiments and evaluation took approximately 48 hours.

## C.1   Agent-Only

### C.1.1   Triangle

Episodes lasted 50 timesteps; training was conducted over 10,000 episodes. Models were updated every 2 episodes, using a batch size of 1,024, sampled from the replay buffer.

Training took approximately 5 minutes.

### C.1.2   9 Points

Episodes lasted 50 timesteps; training was conducted over 5,000 episodes in all scenarios. Models were updated every 2 episodes, using a batch size of 1,024, sampled from the replay buffer.

In the case of $c = 3$, wherein the one-hot agents could only use 3 tokens to communicate about 9 possible targets, we observed some unstable training behavior: agents failed to consistently reach and then stay at the high mean rewards sometimes observed during training. Experiments with different

learning rates and values of $\tau$ over three orders of magnitude failed to resolve this issue. Because this undesirable behavior only arose in this specific scenario of under-parametrized communication, we were not concerned by the bad performance. In order to calculate fair baseline values for one-hot communication, we used the best-performing model from each training run, with models saved every 1000 episodes.

Training took approximately 5 minutes.

### C.1.3 Uniform

Episodes lasted 100 timesteps; training was conducted over 5,000 episodes. Models were updated every episode; using a batch size of 1,024 sampled from the replay buffer of length $100,000$.

In experiments using continuous or prototype-based communication, we used $c = 3$ and $\sigma = 0$; for one-hot communication, we used $c = 5$ to allow for 5 tokens. Initial experiments with other values of $\sigma$ at $0.01$ or $0.05$ showed no large differences in behavior.

Training took approximately 10 minutes.

The interpolation results in Figure 4 were generated by calculating the mean end location of the seeker agent at 10% interpolation intervals for each pair of prototypes. Mean locations were calculated over 20 trials for each interpolation level and pair of tokens.

### C.2 Human-Agent Experiments

In all human-agent experiments, agents were trained in the 2-timestep reference game. In training, models were updated every 50 episodes, using batch size 1,024 from the replay buffer of length 10,000. We used a learning rate of 0.01, and a $\tau = 0.0001$.

Rather than feed agents the direct pixels from images, we pre-trained a CIFAR10 image classifier, based on the SimpleDLA architecture [39]. Trained over 200 epochs with an SGD optimizer with learning rate 0.1, momentum 0.9, and weight decay of $5e - 4$, using random cropping and horizontal flipping of images in training, it achieved 95% accuracy. We used the penultimate layer of the network as a feature extractor; the many-dimensional features were reduced to 10D via principle component analysis (PCA) conducted on the entire training set. Although applying the PCA transform potentially discarded important information for classification, in practice we found that it was sufficient to enable high task performance.

### C.2.1 Zero-Shot Agent Understanding

Training was conducted over 20,000 episodes. The speaker agent used fixed tokens, as explained in Section 3.4. The speaker agent was pretrained for 5,000 epochs on the 100 elements of the supervised dataset, with early stopping with patience 50. A subsequent experiment with 1,000 supervised datapoints (Appendix D) showed no significant change in performance. For the continuous and prototype-based methods, we used $c = 3$; for one-hot, we used $c = 10$. Communication was corrupted with zero-mean Gaussian noise with $\sigma = 0.05$.

Agents using all methods appeared to achieve nearly perfect reference accuracy in training with these parameters, indicating that they had achieved a policy optimum that other hyperparameters would not be able to outperform. Training took approximately 90 seconds.

When using embeddings of AMT-generated communication, we randomly selected any caption from the set of captions generated for images of the true class of the image. This likely underestimated the rich discriminative power of the AMT descriptions. However, given that the AMT captions were generated for images in the CIFAR10 training set, and we evaluated agents with images from the test set, there did not exist an exact match of images and captions.

### C.2.2 Zero-Shot Human Understanding of Tokens

Training was conducted over 50,000 episodes. The speaker agent did not use fixed tokens or grounding data, as we wished to study if emergent communication tokens were human-interpretable. The environment matched that of the previous reference game, but with $\sigma = 0.2$.

**Table 4:** CIFAR10 zero-shot results when training agents with 1000 supervised communication vectors. Results were similar to using 100 supervised examples.

| Method | Label In | Label Out | Self In | Self Out | AMT In | AMT Out |
|---|---|---|---|---|---|---|
| Cont (BERT) | 92% (0.7) | 55% (0.9) | 93% (0.7) | 67% (1.0) | 72% (0.3) | 52% (0.6) |
| Proto (BERT) | 94% (0.5) | 57% (1.0) | 92% (0.8) | 67% (1.0) | 75% (0.3) | 57% (0.5) |
| Disc | 98% (0.2) | 50% (0.4) | 96% (0.4) | 68% (0.9) | | |

**Table 5:** Ablation study for prototype-based communication in the CIFAR10 reference game, using 100 supervised examples. Using both noise and reward enabled the best zero-shot understanding of communication generated from natural language.

| Noise | $R \neq I$ | Label In | Label Out | Self In | Self Out | AMT In | AMT Out |
|---|---|---|---|---|---|---|---|
| Yes | Yes | 93% (0.7) | **58%** (1.0) | 93% (0.7) | 66% (0.9) | 70% (0.4) | **56% (0.6)** |
| Yes | No | 96% (0.4) | 53% (1.0) | 96% (0.4) | 68% (1.0) | 76% (0.2) | 54% (0.7) |
| No | Yes | 91% (0.7) | 55% (1.0) | 91% (0.7) | 65% (1.0) | 71% (0.4) | 54% (0.5) |
| No | No | 96% (0.4) | 53% (0.8) | 96% (0.4) | 67% (1.0) | 74% (0.3) | 53% (0.5) |

Without supervised data, learning successful communication protocols in such settings is challenging [8]. We overcame this difficulty by creating supervised data for the penultimate, softmax layer in prototype speaker agents. That is, we added a crossentropy loss term to agent training to encourage prototype agents to produce a class-specific one-hot vector for images. As this one-hot vector was then multiplied by the token matrix, $T$, which was not subject to supervised data losses, the form of communication remained unguided.

The "communication-space" diagrams, an example of which is shown in Figure 8, displayed 8 2D communication tokens with class labels. Class labels were generated by finding the most common token that the speaker emitted for each image class over 1,000 evaluation runs. The two excluded classes always consisted of one animal and one vehicle; one of those was then used to create the "zero-shot" token that participants had to interpret. Participants were then shown two images, one of each held-out class, and were marked as correct if they selected the image that generated the token. We recorded measured mean participant accuracy. Full details of the user study are included in Appendix F.

Training took approximately 4 minutes.

# D   Appendix: Reference Game Results

In addition to the main results on zero-shot understanding in the reference game results, we conducted additional experiments to assess the importance of different hyperparameters used in training our agents.

First, in Table 4, we repeated the zero-shot experiments, this time training agents with 1000, instead of 100, examples of input images and corresponding tokens. These additional supervised data had a minimal effect on task performance, indicating general similarity of behavior over a large range of supervised dataset sizes.

Second, we conducted ablation studies to assess the importance of the reward matrix and noise in the reference game environment. Results from these studies are included in Table 5. We found that both components were useful in training agents with the best zero-shot understanding of human-generated labels (as shown in the Label Out and AMT Out columns). The results align well with our theory of what enables zero-shot understanding: noise is necessary to induce agents to learn a space of communication instead of just tokens, and a non-identity reward matrix is needed to induce semantic relationships between tokens.

Lastly, we assessed how the agents responded to different natural language inputs at test time. Using the models presented in the main paper, we evaluated their behavior when using embeddings of "animal" or "vehicle" instead of embeddings of the class names.

**Table 6:** CIFAR10 zero-shot reference game mean reward (standard error) using different NLP inputs for agents trained with 100 supervised examples. Using vehicle/animal labels ("binary" rows) worsened in-distribution performance but had no or a small positive effect on zero-shot performance.

| Method | Label In | Label Out |
|---|---|---|
| Cont (BERT) | 91% (0.7) | 55% (1.0) |
| Cont (BERT) - binary | 59% (0.2) | 56% (1.0) |
| Proto (BERT) | 93% (0.7) | 58% (1.0) |
| Proto (BERT) - binary | 61% (0.3) | 60% (1.0) |
| Proto (Hand) | 75% (1.1) | 62% (1.2) |
| One-hot | 96% (0.2) | 50% (0.8) |

**Table 7:** 3D vectors used as hand-crafted word embeddings. These embeddings reflect the desired structure of creating clusters by animals and vehicles.

| Label | x | y | z |
|---|---|---|---|
| plane | 0.25 | 0.9 | 0.35 |
| car | 0.25 | 0.9 | 0.45 |
| truck | 0.25 | 0.9 | 0.55 |
| ship | 0.25 | 0.9 | 0.65 |
| bird | 0.75 | 0.05 | 0.2 |
| frog | 0.75 | 0.25 | 0.2 |
| cat | 0.75 | 0.05 | 0.3 |
| dog | 0.75 | 0.25 | 0.3 |
| deer | 0.75 | 0.05 | 0.4 |
| horse | 0.75 | 0.25 | 0.4 |

Results of these experiments are presented in Table 6. Unsurprisingly, in-distribution performance worsened for both *cont* and prototype models. Not only were embeddings for "vehicle" and "animal" outside the training distribution, but also such labels were poor choices for distinguishing between two animals, for example. However, we observed a small positive effect when using the new labels in out-of-distribution evaluation. This reinforces the conclusion that the *cont* and prototype models learned a communication space that reflected similarities among animals and among vehicles.

## E   Appendix: NLP Embeddings

We used a pre-trained BERT-based English sentence embedder developed by huggingface to convert natural language to embeddings [33]. By default, the embedding dimension is 768; we wished, however, to used a small communication dimension in order to force semantic relationships between the 10 classes. We therefore extracted the raw embeddings for 68 sentences (combinations of class names and stems (e.g., ["Pick the ", "The "] $\times$ ["cat", "dog", "ship"]) and performed 3D PCA. Note that PCA was not informed by the classification task, so embeddings for different classes could have been projected into similar spaces. In many ways, this mimics the dimensionality reduction employed for image feature extraction (explained in Appendix C).

In practice, as demonstrated by Figure 5, even 2D PCA embeddings of the natural language phrases remained informative of class; suggesting that 3D PCA was at least as informative. Furthermore, high-level patterns, such embeddings for words for vehicles clustering together ("plane," "car,", "ship," and "truck" on the right of the diagram), also emerged.

Having fit a PCA transform to the data, we saved the transformation function but discarded the 68 sentences that were used in creating the transform. Instead, when a natural language command was embedded, we started with the natural language, embedded it using our BERT model, and then projected into 3D using the fixed PCA.

Lastly, in addition to the BERT-based embeddings, we also generated hand-crafted embeddings, specifically designed to encourage agents to learn similarities among animals and among vehicles. The embedding values are shown in Table 7. These embeddings were certainly not optimal for the reference game, because the proximity of tokens for different classes led to a high rate of confusion

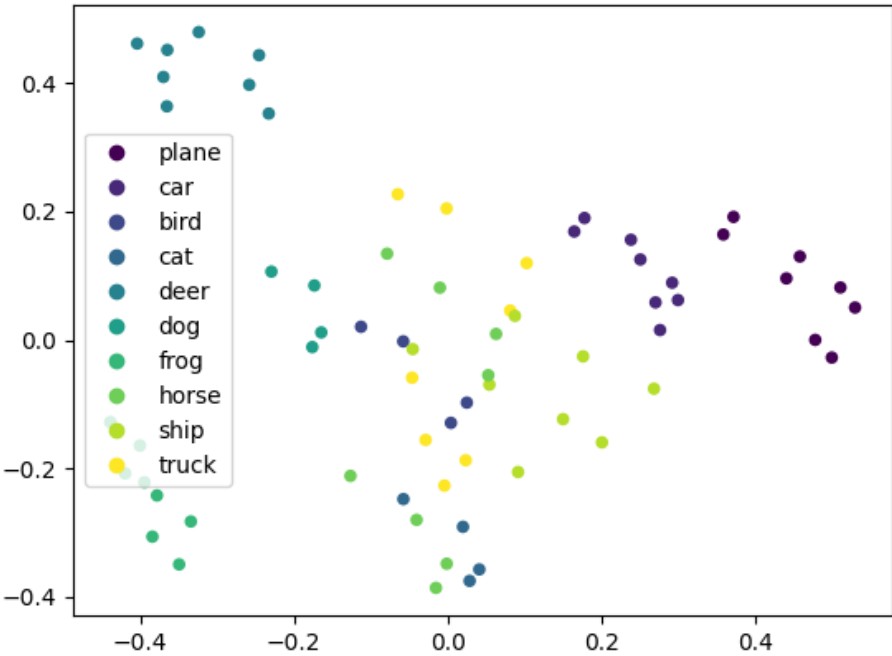

**Figure 5:** 2D PCA of embeddings used to create the low-dimensional language embedder. Even in 2D and without task-specific guidance, language describing the same objects tended to cluster.

due to enviroment noise. However, by clustering tokens to separate vehicles and animals, this pattern did enable better zero-shot performance than our BERT-based models.

## F    Appendix: User Study Details

We recruited participants for online studies, conducted on Amazon Mechanical Turk (AMT), to generate descriptions of images or to interpret emergent communication, as explained in Section 6. This experiment was approved by the university Institutional Review Board (IRB).

In the first data-gathering experiments, we asked participants to generate three labels for images taken from the CIFAR10 dataset. The instruction and user interface of this experiment is shown in Fig. 7. We randomly selected 50 images from each of 10 classes and assigned those 500 samples to workers on AMT. In total, 51 unique workers completed the task, resulting in 1257 valid annotations (some images are not distinguishable due to low resolution). Participants were paid $0.05 for each task; the average completion time was 42 seconds, equivalent to a $4.28 hourly wage.

By asking for 3 labels for each image, we hoped to elicit a diverse dataset beyond merely class labels. As shown in the word cloud visualization in Figure 6, annotations generated by MTurkers include both synonyms of class labels (e.g., dog, puppy, canine) and features of specific images (e.g., furry, small, white). In addition, we note that, due to the low resolution of images in the dataset, some participants incorrectly classified images: e.g., labeling a horse as an ostrich.

In the communication interpretation experiment, participants were asked to distinguish the target image from a distractor based on the zero-shot communication sent by agents. In other words, human participants were playing the role of listener in a reference game, teaming up with trained agents. MTurkers were divided into several between-subject groups in which participants received communications from either our prototype-based models or BERT-based embeddings. We tested 5 different trained models using our method and 1 BERT-based embedding on 80 examples of each of

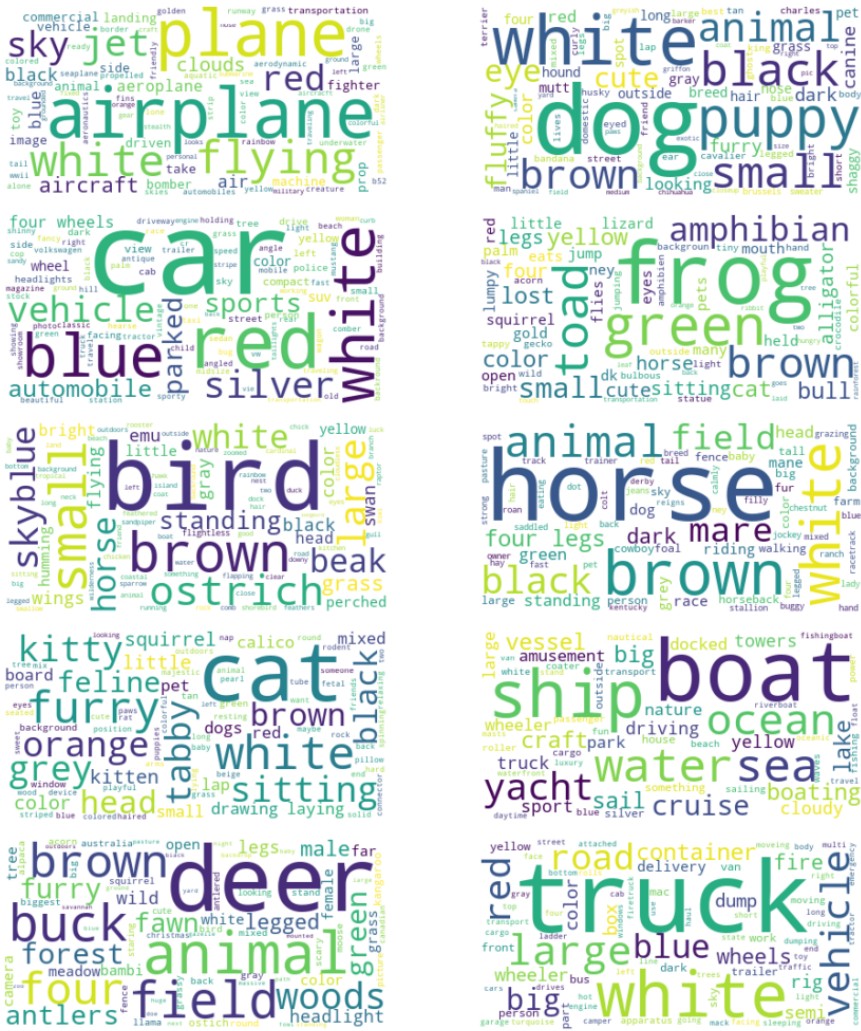

**Figure 6:** Word cloud visualizations of human-generated annotations for images from the CIFAR-10 dataset.

| Group | Samples | Unique workers | Accuracy |
|-------|---------|----------------|----------|
| BERT | 597 | 55 | 65.83% |
| Proto-1 | 294 | 36 | 49.32% |
| Proto-2 | 301 | 47 | 76.08% |
| Proto-3 | 296 | 39 | 70.27% |
| Proto-4 | 300 | 38 | 79.00% |
| Proto-5 | 305 | 38 | 74.10% |

**Table 8:** Performance table of human communication interpretation experiment.

4 animal-vehicle image pairs. After removing tasks that were completed in extremely short or long time (out of 3 standard deviations), we were left with 2093 valid samples from 253 unique workers. Participants were paid \$0.03 for each task; the average completion time was $58.9$ seconds, equivalent to a \$1.84 hourly wage.

The instruction and user interface of this experiment is shown in Fig. 8. Communications were presented in a 2D plane with 8 labeled nodes and one unlabeled communication node. (Labels were generated by evaluating agents in self-play and selecting the most likely token for each class.) Participants were asked to select one out of two images that the communication node most likely referred to. Both images are from the held-out classes, meaning there were no labeled nodes for the

**Examples:**

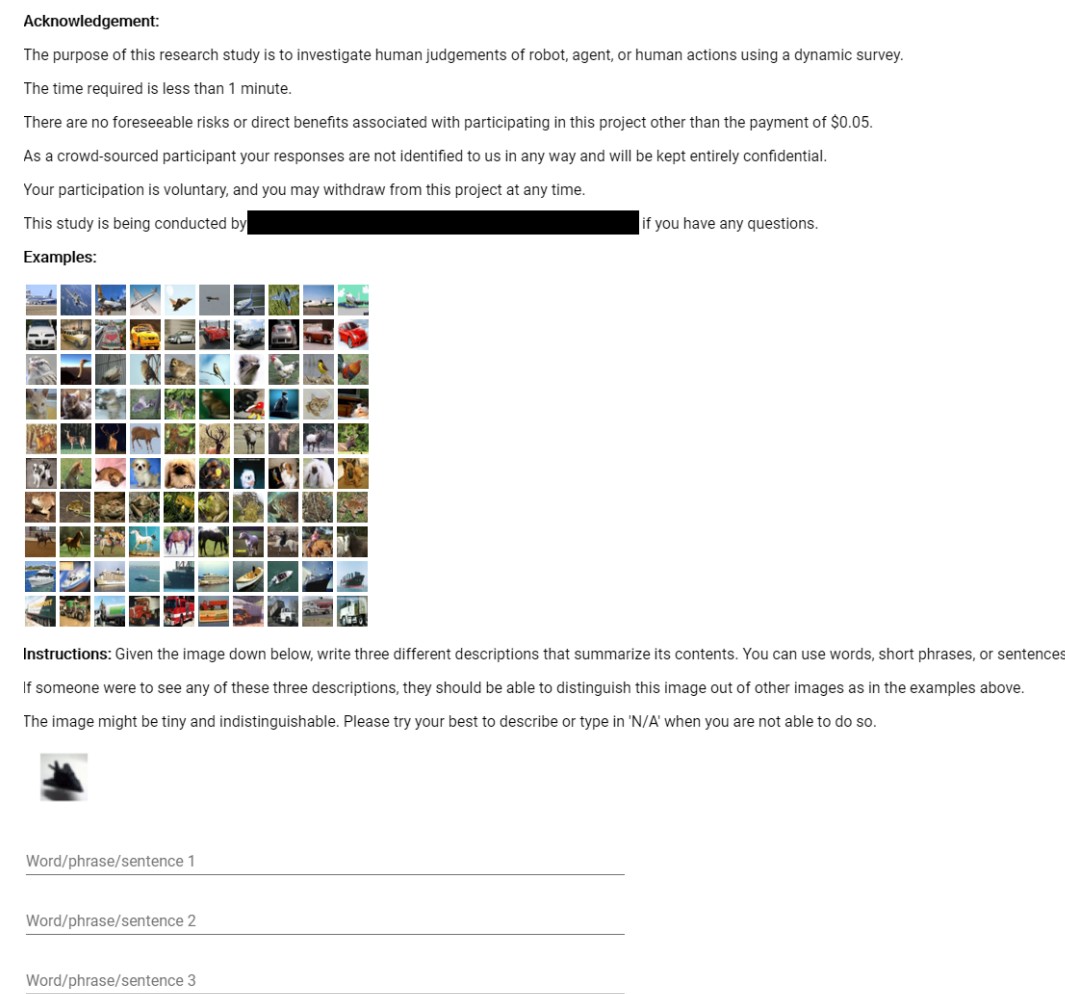

**Instructions:** Given the image down below, write three different descriptions that summarize its contents. You can use words, short phrases, or sentences.

If someone were to see any of these three descriptions, they should be able to distinguish this image out of other images as in the examples above.

The image might be tiny and indistinguishable. Please try your best to describe or type in 'N/A' when you are not able to do so.

Word/phrase/sentence 1

Word/phrase/sentence 2

Word/phrase/sentence 3

**Submit**

**Figure 7:** Instructions and user interface of AMT data-gathering experiment.

image classes, in order to test zero-shot understanding in human-agent teams. As shown in Table 8, 4 out of the 5 trained models using our prototype-based method outperformed the BERT-based embeddings. The difference between two methods is marginally significant ($\chi^2(1, N = 2093) = 3.21, p = .073$). Further inspection of Proto-1, the first trained model that exhibits random-chance performance, revealed that the model failed to converge to high reward in training, and that the tokens for vehicles and animals failed to separate. Training instability is a chronic problem in MARL [8], so we consider this failure as a symptom of general difficulties with reinforcement learning rather than our technique specifically. Nevertheless, in four of our five trials, models did converge to high performance, and if one discards results from Proto-1 as outliers, we significantly outperformed BERT embeddings ($\chi^2(1, N = 1799) = 16.15, p < .0001$).

**Acknowledgement:**

The purpose of this research study is to investigate human judgements of robot, agent, or human actions using a dynamic survey.

The time required is less than 1 minute.

There are no foreseeable risks or direct benefits associated with participating in this project other than the payment of $0.05.

As a crowd-sourced participant your responses are not identified to us in any way and will be kept entirely confidential.

Your participation is voluntary, and you may withdraw from this project at any time.

This study is being conducted by ███████████████████████████ if you have any questions.

**Instructions:**

Your task is to choose a target image out of two candidate images based on a communication map.

The red node is the communication you receive. To give you context, we've labeled 8 blue dots with the types of images those nodes refer to.

Please select which of the two images you think the red node most likely refers to.

Communication map:

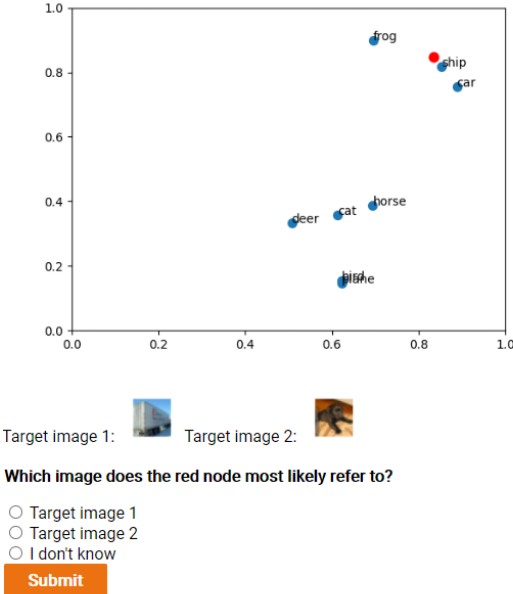

Target image 1:   Target image 2:

**Which image does the red node most likely refer to?**

○ Target image 1
○ Target image 2
○ I don't know

[Submit]

**Figure 8:** Instructions and user interface of AMT communication interpretation experiment.