# OpenReview forum: "Emergent Discrete Communication in Semantic Spaces"
_NeurIPS.cc/2021/Conference — NeurIPS 2021 Poster_

### Official Review · Reviewer_SUhS · 2021-07-12

**Rating:** 7
**Confidence:** 3

**Summary:**

The authors extend previous work on communication in multi-agent reinforcement learning in 2-agent games involving a speaker and a listener, where the speaker generates a token and the listener performs an action informed by the token it receives. The contribution of the present paper is to replace the usual one-hot encoding of tokens with learned tokens in a richer, emergent encoding space. A mathematical justification for this step is given, and its advantages are evaluated in a variety of settings, including settings where humans take the part of either listeners or speakers. These experiments, which show interpolation and zero-shot generalisation, vindicate three hypotheses the authors set out concerning the advantages of the proposed learned encodings over one-hot encodings.

**Limitations And Societal Impact:**

In several of the experiments, the "continual" encoding method out-performs the "prototype" method. Perhaps the authors could discuss this.

**Main Review:**

The paper offers a well motivated contribution to the sub-field of multi-agent RL agent communication, and presents a solid evaluation of the resulting model that demonstrates its advantages. The paper is mostly well written and clear. Overall the paper looks good to me, and deserves to be accepted.

For me, section 3.2 (the mathematical motivation) didn’t add very much to the paper, and the main findings stand for themselves. But other readers might take something useful away from the maths here.

I have a few questions and minor issues.

The authors describe their method as using "prototypes". But what exactly is a "prototype"? Is this the same as a learned token (so there are z prototypes)? I assume this is the case. But then “prototype” seems like a strange word to choose, and I'm not sure what the intuition is. Either way, the term isn’t defined anywhere, as far as I can see. Maybe the authors could remedy this.

In Tables 1 and 2, why do you report the median rather than the mean?

Line 200: “Agents were trained and tested in fixed teams” - Presumably a “team” is a pair of agents: a listener and a speaker? Maybe the authors should make this explicit.

The nature of the listener agent in the particle environment could be described briefly in the main text (if there's space). I gather from the Appendix that it moves according to (somewhat) realistic physics (as opposed to making discrete steps in a grid world). Do its actions comprise a continuously-valued force-direction pair?

Words in human language don’t typically cluster according to semantic in the way advocated here (the “arbitrariness of the sign”). Perhaps, if there's space in the final paper, the authors could discuss this issue. Maybe the richer encodings here are analogous to multiple words rather than single words? (Eg: "fluffy animal" and "dangerous animal" are both semantically close and lexically similar.)

TYPOS

Line 139: upper-case theta shoud be lower-case

Equation 5: Should T_{c x z} be T_{z x c} as in line 138 of the text? Or is it the text that has them the wrong way around? (I don’t think you need those subscripts in the equation anyway)

Line 331: “is essence” -> “in essence”

**Time Spent Reviewing:**

6

---

> ### Author Response · Authors · 2021-08-09
> **Minor clarifications**
>
> We thank the reviewer for their time and address their comments individually below.
>
> 1) Why “prototype” vs. “token”?\
> Indeed, in this paper, prototype is merely a synonym for token, when produced by a prototype-based model. We will clarify the naming convention. In essence, we arrived upon “prototype” from the field of prototype-based classification, in which a discrete set of learnable prototypes are used in a continuous space to classify inputs. Similarly, we initially imagined a set of learnable prototype tokens that communication clusters around. We will clarify the naming in this work and thank the reviewer for catching this issue.
> Tables 1 and 2, median vs. mean? For relatively small datasets, such as the 5 runs for Tables 1 and 2, we favor robust statistics that discard outliers, such as the median. Thus, we intentionally chose the median as we think it is the right sort of statistic to report in these settings, and we wish for the practice to spread.
> 2) Line 200 clarification of “team”?\
> You are correct, a team is a speaker and a listener. We will clarify this point.
> 3) Listener agent dynamics?\
> You are correct that the listener agent moves according to a continuous physics model, double-integrating force, with a friction coefficient. The listener’s action is therefore a two-dimensional, real-valued vector, bounded between -1 and 1 in each dimension. We will add a note to this effect in the main text, in addition to Appendix B.
> 4) Human language clustering?\
> This point is particularly interesting and merits further study. Certainly, human language does not always obey the clustering we motivate and find in our work. For example, in French, “dessus” and “dessous” are antonyms that sound nearly identical to most non-French speakers. It would be fascinating to identify when human language deviates or conforms to our agents’ types of clustering (briefly, we hypothesize that safety-critical or noisy communication settings may produce this clustering among humans, who may exaggerate pronunciation or add distinguishing words like “au dessus”). As we extend this work into more complex domains, we will investigate whether tokens better represent words or general semantic meanings.
>
> Thank you for identifying the typos. In particular, the T matrix should be changed to be c by z, as that corresponds to z column-matrices in c-dimensional space.

---

> > ### Comment · Reviewer_SUhS · 2021-08-25
> > **Response to response**
> >
> > Many thanks for the clarifications

---

### Official Review · Reviewer_AvnV · 2021-07-13

**Rating:** 8
**Confidence:** 5

**Summary:**

The authors present theoretical and experimental results that support the use of employing discrete tokens derived from continuous word embeddings in place of the standard practice of using one-hot vectors as symbols ('words') in emergent communication. The new approach is evaluated using particle-world environments (Lowe et al 2017), in terms of zero-shot reward in a standard referential game on CIFAR10, and in human evaluation. The main contributions are 1) a practical implementation of the technically sound idea of using discretised embeddings to reflect semantic similarities in language, instead of orthogonal one-hot vectors 2) new theoretical results on the optimality of one-hot tokens as vocabulary, and a theoretical analysis of the cases when one-hot tokens might be suboptimal (when the cost of errors or prior over tokens are not uniform, or when the number of tokens and communication dimensions are not equal) 3) experimental results of using those embedding-based tokens, in terms of robustness in noisy environments, interpretability and zero-shot generalization in agent-to-agent and human-to-agent communication.  The authors examine two behaviour shaping constraints that can be used to learn those tokens: 'lexicon-setting' (manually specified and non-trainable tokens) and token grounding using a small supervised set.

**Limitations And Societal Impact:**

The paper lacks the Broader Impact section. Suggestions on how the authors can comment on the broader and societal impact to strengthen their paper:

Discuss the examples of real-life applications of agent-to-agent communication and human-to-agent communication, and the benefits of human-interpretable protocols (refer to the Section 6.2 Zero-Shot Human Understanding of Tokens: the token-based approach achieves higher results than a BERT-based model, which is widely used).

Regarding social impact, the authors could discuss the possible dangers of using the work of human annotators on Amazon Mechanical Turk (AMT) (Section 6.1.).

The authors briefly commented on the biases found in the existing word embeddings, in the future work paragraph. This point could be developed in the Broader Impact section to strengthen the paper.



**Main Review:**

## Strengths:

The authors expand on the existing work on bridging emergent communication and NLP (Lazaridou et al 2020) and propose a sound implementation of word embeddings in multi-agent communication with a discrete channel.

The results are promising in terms of practical human-agent communication.

The tokens are shown to be meaningfully related (e.g, a mean of the tokens representing locations $\alpha$ and $\beta$ represents the actual mean of the two locations on a grid), which opens new possibilities in terms of interpretable communication protocols and zero-shot generalization in multi-agent tasks.

The authors include anonymised code in the submission.

In terms of writing and structure, the submission is significantly above average: for example, the authors clearly state the hypotheses which motivate their experiments. The appendix is detailed and the figures are of (above average) quality. Well done!

The authors motivated their answers in the Checklist.

The writing is quite dense in terms of the amount of technical content, but it is clear and coherent, with no unjustified claims.

## Originality:

The contributions are original and the previous work in multi-agent communication is appropriately cited.

## Quality:

As mentioned in the Strengths section, both the content and the presentation are of high quality.

## Questions to the authors:

In Table 3, how many runs did you use to compute the mean reward and standard errors?

Have you observed structural, semantic or pragmatic language drift (following Lazaridou et al 2020) in your experiments?

## Significance:

The paper proposes a way of deriving a semantically interconnected yet discrete vocabulary in multi-agent communication, which might influence the research on discrete, interpretable communication channels. Zero-shot generalization, interpetability and various means of incorporating a priori knowledge in the agents' protocol based on the task are of interest to the research communities (emergent communication, deep RL) working on multi-agent communication.

## Notation:

Line 108: ‘ in a matrix of $z$ tokens’ suggests that $z$ is a number whereas in Equations (1)-(4) it is used as a set ($i \in z$). Please change the limits of the sums and the maxima to something like $1 \leq i \leq z$.

Line 124: The usual notation for variance is $\sigma^2$, $\sigma$ suggests standard deviation.

## Typos:

Line 7: ‘architectures that enables’ -> ‘architectures that enable’

Line 12: ‘allowing them communicate’ -> ‘allowing them to communicate’

Line 138: ‘$T_{z \times c}$’ -> ‘$T_{c \times z}$’

Line 244 and line 245: a hyphen should be replaced with a minus -> $-100$, $-96$

## Latex:

Line 93 $\forall i \in [1,N]$, $A_i$ is a set of actions

'...' in equations should be replaced with '$\ldots$'

## Linguistic comments:

Line 117 ‘We seek to find’ -> ‘We seek’

Line 138 ‘which then is multiplied’ -> ‘which is then multiplied’




**Time Spent Reviewing:**

11 hours

---

> ### Author Response · Authors · 2021-08-09
> **Minor clarifications; general appreciation for guidance**
>
> We thank the reviewer for their particularly insightful feedback. Their detailed comments will be incorporated to improve the readability of the paper. The more general feedback on impact (comparison to wide-spread BERT-based embeddings, troubles with biases in embeddings, etc.) is quite interesting and motivates future work.
>
> **Response to Questions:**
> 1) In Table 3, how many runs did you use to compute the mean reward and standard errors?\
> We apologize for the lack of clarity. We trained 120 speaker-listener teams (5 teams for each of the 24 possible animal-vehicle combinations). For self-play and label-based experiments, each team was evaluated with 500 trials. For AMT-based experiments, we used 1,257 annotations produced by our surveys. This information is present in the paper (the first and second paragraphs of Section 6.1 and Appendix F), we will consolidate this information and present it within the table caption.
> 2) Have you observed structural, semantic or pragmatic language drift (following Lazaridou et al 2020) in your experiments?\
> We were cognizant of these risks, due to the prior art alluded to. This was a particular concern for human-agent experiments, in which the agent communication needed to remain aligned with human-interpretable tokens. We observed drift among tokens if we turned off the supervised loss (i.e., the mean squared error loss grounding inputs to particular tokens) in the middle of training. However, by maintaining supervised loss throughout the RL training, we appeared to avoid obvious drift issues. This is, however, a promising question for future analysis of this work.
>
> If space allows, we would certainly like to include more analysis of broader impacts, including possible uses of the technology (compared to BERT) and data generation (from AMT). Particularly, our proposed method is applicable to RL agents in human-agent teams where natural and efficient communication plays an important role in maintaining shared situation awareness among team members. The zero-shot human/agent understanding experiment is a simulation of novel task context or team reorganization in real life teamwork. In such cases, team members do not have the chance to reach an agreement on the common semantics of communication tokens prior to the task (e.g. co-training), rather, they need to understand novel tokens in a zero-shot fashion. Human interpretable communications enable a faster convergence of team state by having humans a shorter learning phase to understand agent’s communication.
>
> As for the data generation process on AMT, one potential risk is the prior exposure to CIFAR materials of the AMT user group. Since CIFAR image dataset has been widely used in computer vision tasks, participants might be exposed to those images and corresponding true labels previously, resulting in a less noisy human communication in the experiment reported in section 6.1.

---

### Official Review · Reviewer_HBxw · 2021-07-15

**Rating:** 7
**Confidence:** 3

**Summary:**

This paper extends research in multi-agent emergent communication using discrete tokens. The agents learn to communicate across a noisy channel. The main idea is to have the agents learn to embed discrete tokens into a continuous space, similar to NLP techniques such as word2vec. The authors show that the learned tokens are interpretable to other agents in self-play. They also show that these messages can be adapted to understand embeddings produced by BERT or human labels, and that humans can in turn interpret the embeddings.

**Limitations And Societal Impact:**

The authors address the limitations and social impact of this work.

**Main Review:**

Overall, I enjoyed reading this paper. The interpretability the actions and communications of an RL agent is an important problem which the paper clearly and thoroughly studies. The main idea is simple to understand and well motivated. The results are in simple settings.

Pros:
1) Equations are explained clearly and the paper is well written.
2) Experimental results show clear improvement over the baselines in the first two environments.
3) Experiments and ablations are thorough. I particularly liked the results on human-interpretability of the agent messages.

Cons:
1) It would be nice to see some agent-only environments beyond particle worlds, and tasks beyond navigating to the correct -landmark. Similarly, it would be interesting to see performance on a slightly larger set of image classes.
2) The prototype agent doesn't seem significantly different from the cont agent on the zero-shot reference game.
3) 100 supervised examples in the training set is not quite the same as zero-shot learning.

Questions:
1) Do the agents communicate once per episode, or continuously? If the latter, do the messages change?
2) What are the one hot agent messages when c < z?
3) In Section 5 Figure 3, why are the final locations only roughly clustered? Given one of five messages, the seeker agent should reach one location and stop to maximize expected reward. Figure 4 seems to support this.
4) In Appendix D, it would be interesting to see the non-identity reward matrix.
5) For the zero-shot human understanding experiments, are the held out classes always 1 animal and 1 vehicle? Is there any interesting pattern in human errors e.g. classes that are conflated consistently? Was the same experiment done with the cont model?

Details:
- (Section 5 Table 1) - Some additional noise ablations would be interesting.
- (Section 5 Table 2) - Some additional experiments varying z would be interesting.
- (Section 6.2 Line 331) - is essence -> in essence.
- (Section 6.2) - Add reference to the results in Appendix F/Table 8. A reference to Figure 8 is also helpful.


**Time Spent Reviewing:**

3

---

> ### Author Response · Authors · 2021-08-09
> **Minor clarifications about zero-shot learning and observed behaviors**
>
> We thank the reviewer and address their comments individually below:
>
> **Addressing Cons:**
>
> We agree with the reviewer’s desire to apply this technique to richer domains; we plan to do so in future work.
>
> It is indeed interesting that cont and prototype agents do not appear to differ significantly in zero-shot settings. On the one hand, the fact that a discrete method like prototypes can match continuous agents is a result in and of itself. On the other hand, in domains that are fundamentally best represented as discrete, one would expect discrete methods to actually outperform continuous methods. Perhaps the distinction between cont and prototype agents would be best clarified in “generalized zero-shot learning” contexts, in which agents must decide when an item belongs to a known category or a novel one. We will pursue this in future work.
>
> The third point listed by the reviewer under “Cons” represents a slight misunderstanding: we use 100 labeled examples to supervise agent training, but those 100 examples do not belong to the two held-out classes. Similarly, agents are never trained, even in self-play, with the held-out classes. Thus, evaluation with images from those classes truly is zero-shot.
>
> **Addressing Questions:**
>
> 1) The agents communicate at every timestep. In theory, messages should not change across timesteps, and in practice, we observed nearly perfect consistency across a given episode. We merely adopted this design to avoid the slightly increased complexity of recurrent agents.
>
> 2) We assume you mean z to represent the minimum number of distinct tokens needed to perform a task optimally. When c < z, one-hot agents are fundamentally under-parametrized as there are fewer actual one-hot tokens available than what would enable optimal behavior. We explore such a scenario in the 9-points environment: in the 3rd row of Table 1, we see that for c = 3, one-hot agents devolved to treating clusters of targets as a single target. Generally, one-hot agents learn a discretization with c tokens, so if c is less than the number of distinct messages needed for optimal behavior, one-hot agents must perform suboptimally. Conversely, by decoupling c and z, our prototype agents sidestep this issue.
>
> 3) Figure 4 represents the mean destination across many runs of the seeker agent; Figure 3, by plotting individual runs, shows the noise across trials. Some clusters are present in Figure 3, but as the reviewer notes, the clusters are not perfect. Part of this noise is simply a product of the random starting location of the seeker agent: if the seeker was spawned far from the target, it may be unable to reach the target by the end of the episode. More generally, though, the uniform environment is not conducive to particularly tight clusters. Because targets were drawn from a continuous space, discrete communication made even the optimal overall communication protocol (very tight clusters) suboptimal for individual episodes. Thus, the signal driving agents towards tight clustering policies became weak once agents started forming at least weak clusters. We observed similar clustering patterns with one-hot agents, indicating that this is a generally hard trait to learn in the environment, rather than a symptom of our technique in particular.
>
> 4) We will explicitly include the matrix in Appendix D. The contents are described in Section 6, line 290-295, in which correct predictions got reward 1, incorrect predictions of the right group got reward 0.75, and incorrect predictions of the wrong group got reward 0.
>
> 5) Yes, all zero-shot human understanding experiments were conducted with one animal and one vehicle. We did not perform similar experiments with the cont model, because it is not obvious what the right “communication map” to show to participants would be. Choosing random messages that resulted in the correct classification would be one approach but feels unprincipled, yet we lack a more formal method for selecting the right set of messages from a continuous space. In fact, this inability to label points in a principled way illustrates one of the advantages of our prototype agents: the vocabulary is immediately obvious.\
> In analyzing participants' abilities to understand unlabeled tokens, we indeed observed interesting patterns. Among all vehicle-animal pairs that we tested with prototype-based communication, "plane-bird" was the most confusing combination for participants. Participants got an accuracy of 46.88% when differentiating images from those two classes, which is significantly lower than the average accuracy of 70%. This aligns with our intuitive understanding that "bird" and "plane" have relatively close semantic meanings. Such patterns of confusion among humans indicates the efficacy of our method in conveying semantic information from agents to humans. As a baseline, human participants did not exhibit a varied confusion matrix when receiving BERT-based communications, for which the accuracy for all pairs was around the mean accuracy (66%).
>
> We thank the reviewer for the detailed comments, and will address them in a final version of the paper. Varying noise, communication dimension, and reward matrices would certainly be interesting.

---

### Official Review · Reviewer_nATd · 2021-07-17

**Rating:** 8
**Confidence:** 2

**Summary:**

The paper argues that using discrete tokens in emergent communication fundamentally limits what agents can learn.

Method: Instead, they propose to learn an embedding matrix in addition to a token selection policy for use in emergent communication. The embeddings are trained in 3 different ways: supervised, learned from scratch, and manual specification. This approach is corroborated by analyzing a noisy channel reference game in an information-theoretic manner.

Their self-play tasks are in a particle-world environment:
- In _triangle_ a speaker must communicate which point of the triangle the listener should go to.
- In _9 points_, a speaker must communicate which point of the triangle the listener should go to. But, the 9 points are grouped into 3 groups of 3. The agent is rewarded based on distance, so going to a point in the same group as the target is less costly.
- In _uniform_, the positions of two landmarks are sampled randomly. A seeker observes the id of the target and an observer observes their locations. They must communicate to guide the seeker to the target.

They find that their proposed discrete prototype model outperforms a one-hot representation. In the _9 point_ task, the embedding space also reflects the point grouping.

In their zero-shot human communication experiments, they train speakers and listeners on a referential game over pairs of CIFAR images. They compare learned embeddings performance against class-label embeddings, and embeddings trained on sentences generated by humans for the task. The aim of these experiments was to show that agents were able to generalize to human communication and to generalize to novel communication settings (ie to images unseen before).

**Ethical Concerns:**

The authors exhibited good general ethical conduct when conducting human experiments.

**Limitations And Societal Impact:**

Yes, they address limitations under the contributions section.

**Main Review:**

Originality: Adding embedding tokens to an emergent communication task is a sensible contribution. The creativity of this paper shines in the way the experiments were designed. E.g., the 3 evaluation tasks were well designed to understand how the agents change communication strategies as imprecision becomes less costly (triangle -> points) and how agents use discrete tokens to reason about continuous phenomenon. Changing out the different kinds of embeddings in the CIFAR experiments was also a clever way of understanding how the communication protocol changes across different tokens.

Quality: the experiments were thorough and well executed (e.g., median and standard error reported, experiment design and rationale was highly detailed.)

Clarity: The paper was very straightforward to follow. From the experimental section, I felt I could reproduce the results. Specifically, there visualizations communicated the key concepts and environments very effectively.

Significance: Adding embeddings as tokens in emergent communication could open up many avenues for analysis going forward. E.g., the paper already demonstrated how embeddings can better demonstrate how humans understand emergent communication protocols and how communication can be transferred to new settings (such as differentiating between new images).


**Time Spent Reviewing:**

2

---

> ### Author Response · Authors · 2021-08-09
> **Thank you for your thorough review**
>
> We thank the reviewer for the time and effort they dedicated to reading the paper carefully. They have clearly understood both the method and experiment design. We look forward to continuing this work in the future.

---

### Decision · Program_Chairs · 2021-09-28

**Decision:**

Accept (Poster)

**Comment:**

This work studies referential games in between a speaker and a listener and proposes to replace the commonly used one-hot encoding of tokens with learned tokens in an encoding space. Moreover, the authors designed a good experimental setup to help them test and understand the effect of their approach and a thorough evaluation, including evaluation with humans.

All reviewers agree that this is a solid contribution to the field of emergent communication, positively commenting not only about the general structure of the paper on putting forward a hypotheses and then thoroughly testing it with targeted experiments, bit also commenting on the creative experimental setup but also the interpretability aspects of their proposed method.

This is a good contribution to the field of emergent communication and hence I suggest acceptance of this work.

**Consistency Experiment:**

NeurIPS has a long history of experimentation. In 2014, NeurIPS ran an experiment in which 10% of submissions were reviewed by two independent committees to quantify the randomness in the review process. This year, we repeated a variant of this experiment to see how the quality of the review process has changed over time.  This paper was part of the experiment and was therefore assigned to two committees (consisting of reviewers, an Area Chair, and a Senior Area Chair) that reached independent decisions.  If both committees made the same recommendation, this recommendation was followed. If a single committee recommended acceptance, the paper was accepted (with the exception of a few cases in which the other committee identified what we considered a fatal flaw, e.g., an error in a key result).

This copy’s committee reached the following decision: **Accept (Poster)**

The other committee assigned to the paper recommended **Reject**.  You can find the other set of reviews, along with any follow up discussion with the authors here:
https://openreview.net/forum?id=hsqZ5v8PFyQ